# Molecular insights of exceptionally photostable electron acceptors for organic photovoltaics

Zhi-Xi Liu[1,6], Zhi-Peng Yu[1,5,6], Ziqiu Shen[1], Chengliang He[1], Tsz-Ki Lau[2], Zeng Chen[3], Haiming Zhu [3], Xinhui Lu [2], Zengqi Xie[4], Hongzheng Chen[1] & Chang-Zhi Li [1✉]

Photo-degradation of organic semiconductors remains as an obstacle preventing their durable practice in optoelectronics. Herein, we disclose that volume-conserving photo-isomerization of a unique series of acceptor-donor-acceptor (A-D-A) non-fullerene acceptors (NFAs) acts as a surrogate towards their subsequent photochemical reaction. Among A-D-A NFAs with fused, semi-fused and non-fused backbones, fully non-fused PTIC, representing one of rare existing samples, exhibits not only excellent photochemical tolerance in aerobic condition, but also efficient performance in solar cells. Along with a series of in-depth investigations, we identify that the structural confinement to inhibit photoisomerization of these unique A-D-A NFAs from molecular level to macroscopic condensed solid helps enhancing the photochemical stabilities of molecules, as well as the corresponding OSCs. Although other reasons associating with the photostabilities of molecules and devices should not excluded, we believe this work provides helpful structure-property information toward new design of stable and efficient photovoltaic molecules and solar cells.

[1] State Key Laboratory of Silicon Materials, MOE Key Laboratory of Macromolecular Synthesis and Functionalization, Department of Polymer Science and Engineering, Zhejiang University, Hangzhou, P. R. China. [2] Department of Physics, The Chinese University of Hong Kong, New Territories, Hong Kong, P. R. China. [3] Department of Chemistry, Zhejiang University, Hangzhou, P. R. China. [4] State Key Laboratory of Luminescent Materials and Devices, School of Materials Science and Engineering, South China University of Technology, Guangzhou, P. R. China. [5] Present address: Institutes of Physical Science and Information Technology, Anhui University, Hefei, P. R. China. [6] These authors contributed equally: Zhi-Xi Liu, Zhi-Peng Yu. ✉email: czli@zju.edu.cn

Power conversion efficiencies (PCEs) of organic solar cells (OSCs)[1–4] are approaching the threshold of practical application[5,6]. However, the photodegradation of organic semiconductors remains to be an obstacle preventing the practice of such optoelectronics[7–10]. Typically, the desirable operational lifetime of OSCs[11–13] is only secured under the exclusion of oxygen and UV light. It is because that the intrinsic photo-stabilities of organic photovoltaic materials, especially for the high-efficiency acceptor–donor–acceptor (A–D–A) non-fullerene acceptors (NFAs) are not ideal yet that could degrade under illumination in ambient[14–16]. In particular, typical A–D–A NFAs are synthesized through Knoevenagel condensation (KC) of aldehyde and active methylene[17–19], wherein the exocyclic vinyl groups are constructed to the conjugate electron-rich core (D) with electron-deficient terminals (A). This structural design enables the excellent tuning of the optoelectronic properties, hence high photovoltaic performance of A–D–A NFAs, and also undermining their intrinsic photochemical stabilities[14,20]. It is because the exocyclic vinyl double bond is chemically more vulnerable toward photooxidation than those in aromatic units[15]. Besides, the vinyl bonds from KC reaction exhibit reversible reactivity and could be hydrolyzed back to starting materials[21]. Therefore, to prevent the side reaction of exocyclic vinyl group will be critical to enhance the intrinsic stability of A–D–A NFAs, which suffers a lack of effective approach so far.

Recently, semi-fused and non-fused ring acceptors (SFRAs and NFRAs) have been developed[16,22–24], which possess relatively low synthetic complexities and decent PCEs. To our surprise, among them, NFRA PTIC exhibits a steady absorption profile even after 32-h one-sun-equivalent illumination in ambient, suggesting a photostable A–D–A acceptor[16]. Notwithstanding all of those efforts, the understanding is quite limited, and there lacks of knowledge on how to access A–D–A NFAs with excellent resistance towards photochemical reaction. It, therefore, motivates researchers to explore organic semiconductors with inherent photochemical stabilities, as well as to understand their structure–activity relationship. These efforts should be highly valuable for eventually developing stable organics and related optoelectronics.

In this work, we investigate the photodegradation pathway of A–D–A NFAs with the representative molecular skeletons of fused, semi-fused and non-fused backbones (Fig. 1a), and disclose that volume-conserving photoisomerization of the exocyclic vinyl group in these unique molecules acts as a surrogate toward their subsequent photochemical reaction, such as photooxidation of A–D–A NFAs under aerobic condition (Fig. 1b, c). This undesired photochemical reaction breaks the conjugation and intramolecular charge transfer of A–D–A NFAs, hence resulting in the irreversible performance decay of non-fullerene OSCs. Exceptionally, we reveal non-fused acceptors, representing one of the rare existing samples, exhibit appearing characteristics of both good photochemical tolerance and photovoltaic performance. The intrinsic photostabilities of NFAs, among the studied samples, are sequenced in the order of non-fused PTIC (degradation rate: $9.00 \times 10^{-4}\,\text{h}^{-1}$) > non-fused PTICH ($2.30 \times 10^{-3}\,\text{h}^{-1}$) > semi-fused HF-PCIC ($1.36 \times 10^{-2}\,\text{h}^{-1}$) > fused IT-4F ($1.48 \times 10^{-2}\,\text{h}^{-1}$), under one-sun-equivalent illumination in ambient. Non-fused PTIC-based OSCs display about 359 times and 322 times slower decay rates than fused IT-4F and semi-fused HF-PCIC-based OSCs, respectively.

More importantly, along with a series of in-depth investigations, we identify that the structural confinement of these A–D–A NFAs from the molecular level to macroscopic condensed solid is particularly important toward enhancing the photostabilities of A–D–A NFAs, as well as the derived non-fullerene OSCs. There are two primary factors for achieving so: (1) to suppress the photoisomerization of vinyl groups by installing outward-chain in

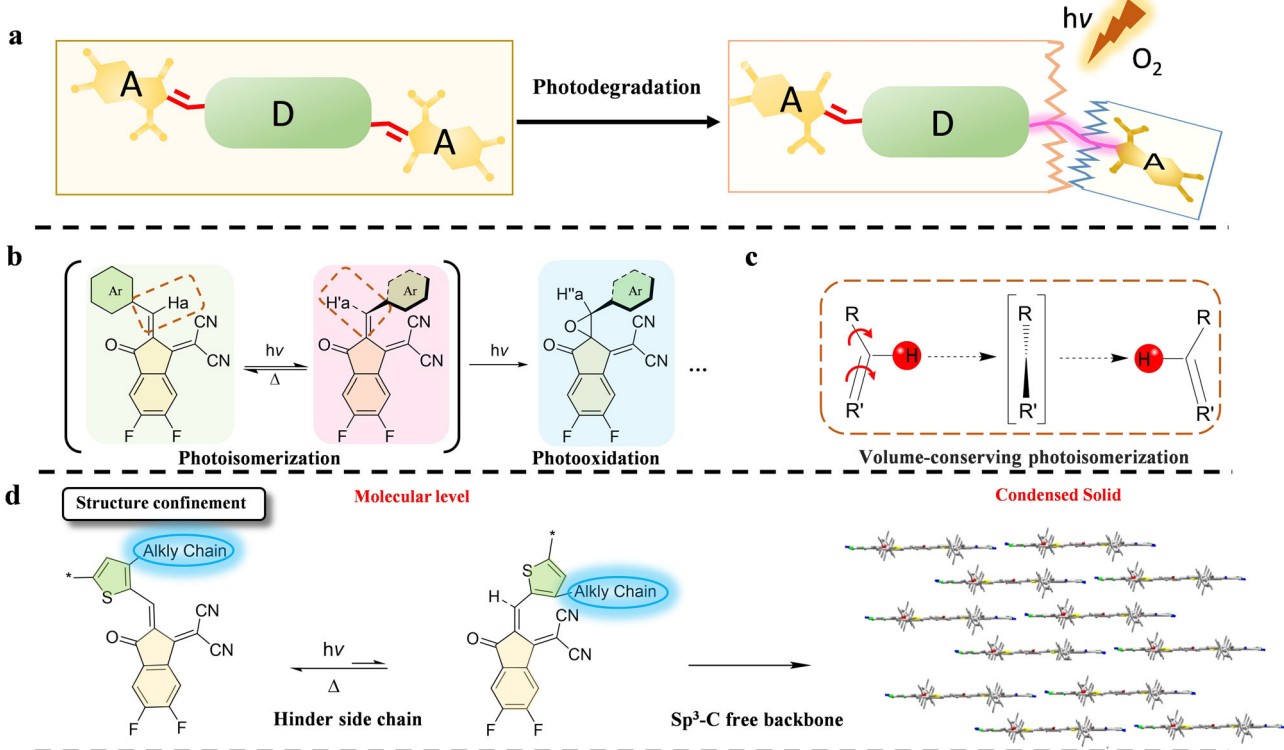

**Fig. 1 Structural factors on determining the photostability of A–D–A NFAs. a** Structural sketch and the photodegradation of typical A–D–A NFAs. **b** Photochemical reaction. **c** The plausible isomerization mechanism of the vinyl group in A–D–A NFAs. **d** Structural confinement from the molecular level to macroscopic condensed solid for enhancing photostabilities of NFAs.

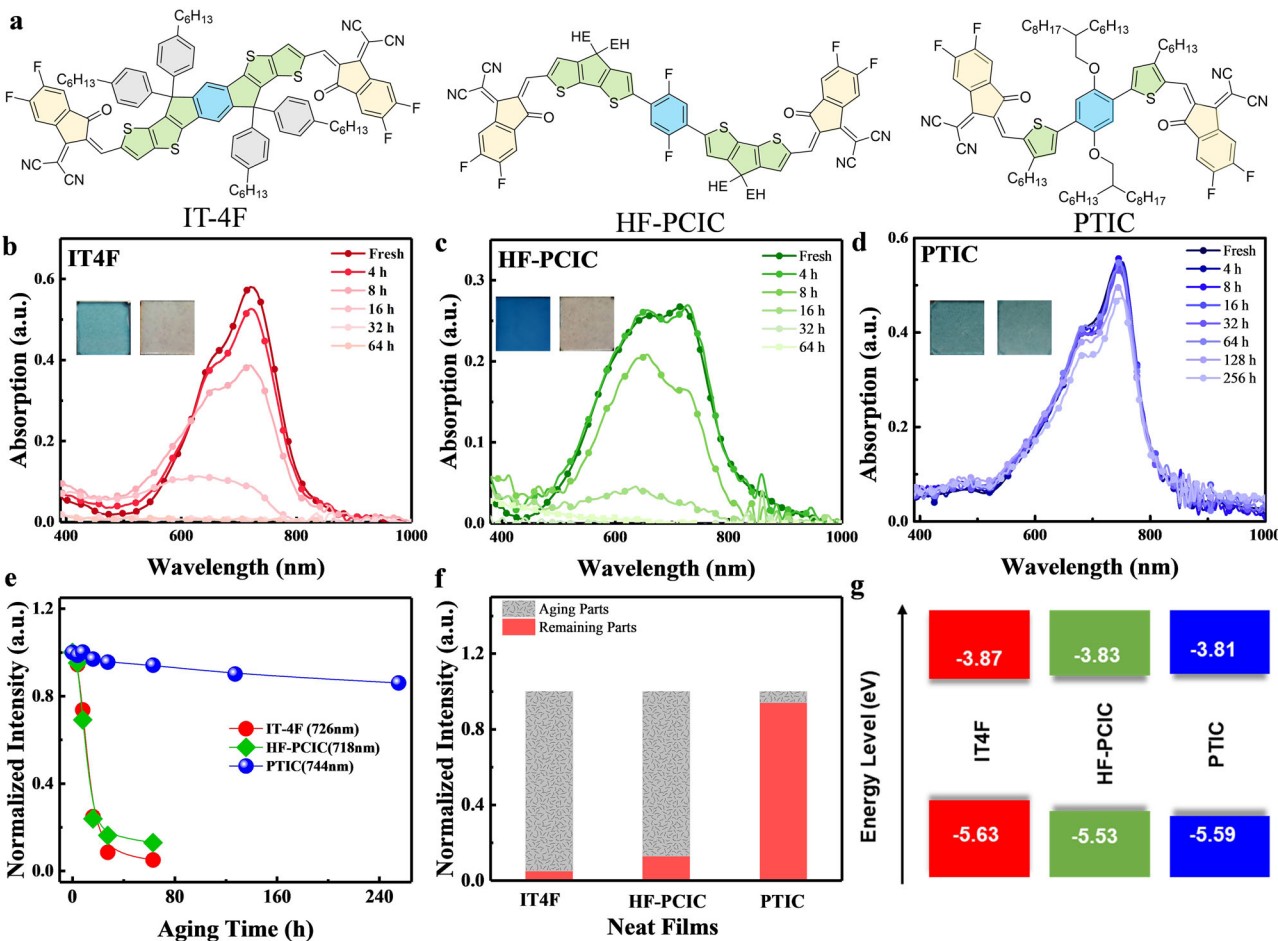

**Fig. 2 Intrinsic photostabilities of A–D–A NFAs. a** Molecular structures of FREA IT-4F, SFRA HF-PCIC, and NFRA PTIC. **b–d** Their film UV–vis absorption spectra and photo images (before and after 64 h) under one-sun-equivalent illumination in ambient. **e** The change and **f**, the percentage of remaining absorption intensity at 64 h illumination. **g** Energy levels for different acceptors. EH 2-ethylhexyl.

A–D–A NFAs (Fig. 1d); (2) to promote dense packing of molecules with planar $Sp^3$ carbon-free backbones (Fig. 1e). As result, NFRA PTIC allows possessing exceptional photostability among a series of the studied examples. And the fused ring acceptor, such as Y6, fitting well with this guidance, has also achieved the enhanced photostability. Even though, there should not exclude other reasons associating with the photostabilities of non-fullerene acceptors and solar cells. We believe the structure–property correlation from these unique examples provides new insights on designing efficient and stable organic semiconductors and the derived solar cells.

## Results and discussion

We select A–D–A NFAs with representative skeletons as studying examples, including the fused (IT-4F), semi-fused (HF-PCIC), and non-fused-cores (PTIC), respectively (Fig. 2a). Their photostabilities are first investigated by recording the UV–vis spectra of neat films under the continuous one-sun-equivalent illumination in ambient conditions (Fig. 2a and Supplementary Figs. 1 and 2). Among them, the photostability of non-fused PTIC is sharply distinct from others. Note that the maximum absorption peaks ($\lambda_{abs}^{max}$) are in the similar range of 726 nm (IT-4F), 718 nm (HF-PCIC), and 744 nm (PTIC), respectively. After illumination of 64 h, the dramatic decay of maximum absorption ($A_{abs}^{max}$) is observed for fused IT-4F and semi-fused HF-PCIC, appearing nearly transparent with less than 15% of the original $A_{abs}^{max}$ due to the photochemical degradation of

molecules (Fig. 2b). In sharp contrast, PTIC film exhibits an excellent photostability, which remains 94% (after 64 h illumination), and 84% of original absorption intensity after 256 h illumination (Fig. 2c, d). The photodegradation rate of neat films is calculated to be $1.48 \times 10^{-2} h^{-1}$ (IT-4F), $1.36 \times 10^{-2} h^{-1}$ (HF-PCIC), and $9.00 \times 10^{-4} h^{-1}$ (PTIC) in the first 64 h illumination. The photostability of PTIC remains distinct as well in the bulk heterojunction blend with polymer donor, PBDB-TF, wherein PTIC maintains absorption characteristics well while the fast decay of polymer (Supplementary Fig. 3). A similar decay tendency has been observed in the encapsulated films (Supplementary Fig. 4), which, again, show PTIC film and PBDB-TF:PTIC blend possess superior photostability. Note that the light spectra we employed in this study contains a high portion of ultraviolet photons (250–380 nm), which is the common source of instability of organic materials that associate chemical bond dissociation under high-energy photon irradiance[25,26]. The photochemical degradation of a conjugated polymer, as electron donors, would involve a chain radical oxidation process, similar to that of P3HT[27]. To obtained more information, photoluminescence (PL) of neat films was measured (Supplementary Fig. 5). PTIC films remain identical PL profiles before and after aging. Whereas, IT-4F film shows a completely different emission profile with significant blue-shift peaks after aging. It would suggest that the degraded species have large shrinkage of π-conjugated aromatic systems, associating with the deterioration of optoelectronic and charge transport properties of photovoltaic materials, hence impacting on the device performance.

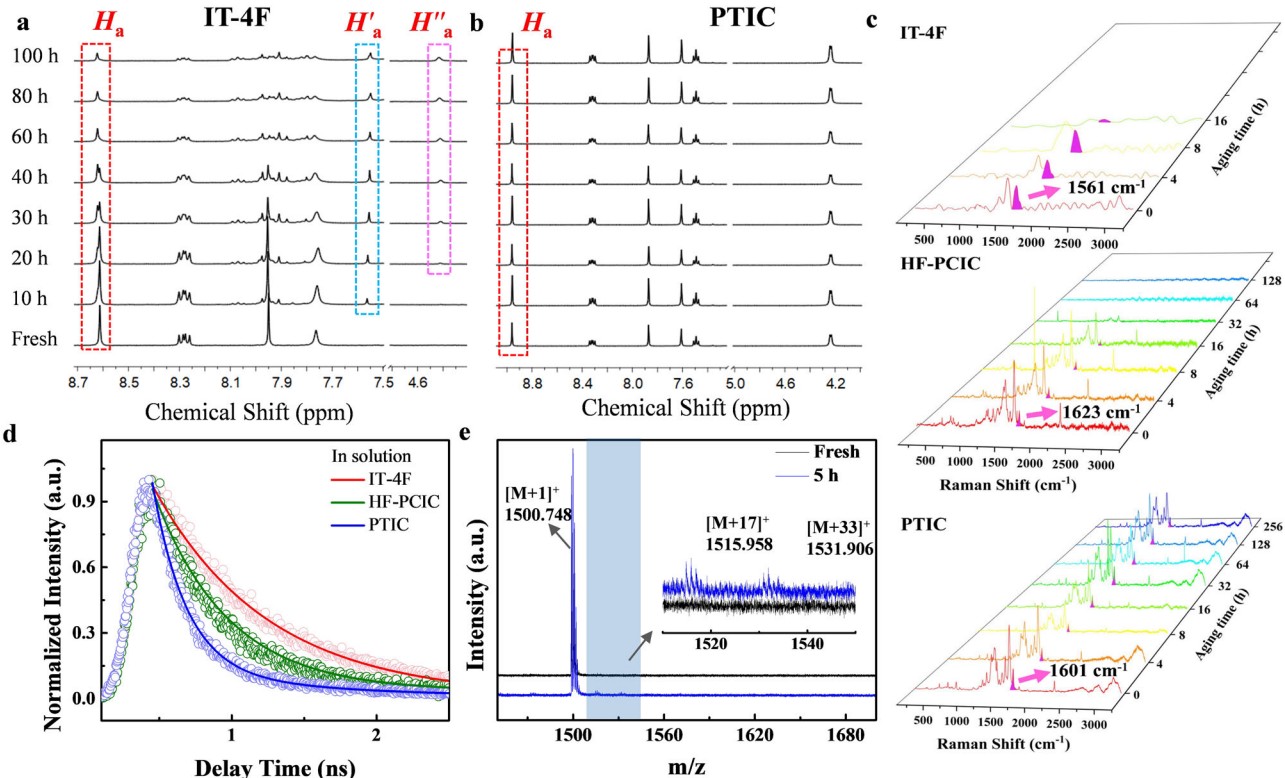

**Fig. 3 Structural characterizations of photo-illuminated NFAs.** [1]H NMR of **a**, IT-4F and **b**, PTIC in solution under continuous one-sun-equivalent illumination. Line 1–8 represents the signals at different times (fresh, 10 h, 20 h, 30 h, 40 h, 60 h, 80 h, 100 h). **c** Raman spectra of IT-4F, HF-PCIC, and PTIC under continuous one-sun-equivalent illumination. The highlighted purple peaks represent the signals of the vinyl bond. **d** Time-resolved photoluminescence for IT-4F, HF-PCIC, and PTIC in solution. **e** MALDI-TOF of IT-4F film before and after illumination in ambient.

Considering these studied NFAs discussed in the text possess similar bandgap and energy levels[28], the varied photostability of different NFAs is ascribed to their structural and stacking properties (Fig. 2e).

To reveal the influence of photostability on their chemical structures, we have monitored the structural change of NFAs by [1]H nuclear magnetic resonance ([1]H NMR) spectra. It reveals that, under illumination, the vinyl groups of IT-4F underwent photoisomerization, and then followed by photooxidation with the generation of clear epoxide species, along with the other decomposed species upon extending illumination time. The original chemical shifts of vinyl protons ($H_a$) are characteristics at 8.55 ppm (singlet for IT-4F), 8.85 ppm (multi peaks for HF-PCIC), and 8.95 ppm (singlet for PTIC) in dichlorobenzene-$d_4$, respectively (Fig. 3a and Supplementary Figs. 4–6). The fused IT-4F shows the distinct structural change with a new proton peak at 7.57 ppm (denoted as $H'_a$) raised up accompanying the decrease of vinyl protons ($H_a$) within the first 10 h illumination (details will be discussed hereafter) (Supplementary Fig. 7–9). Followed by this, a new peak at 4.51 ppm (denoted as $H''_a$ for IT-4F) appears, which is assigned as the proton signal of epoxide[14,15]. Whereas, it is interesting to note that PTIC remains stable in solution even after 220 h illumination (Fig. 3b). The chemical shifts of HF-PCIC also remain little change in solution, whereas its film fades quickly in the neat film. These results also agree well with the thin layer chromatography (TLC) experiments (Supplementary Fig. 12). The reason why PTIC and HF-PCIC are stable in solution is that their unfused backbones allow fast nonradiative relaxation of high-energy excited state[29], versus of those of fused structures (IT-4F) in solution. It is consistent with time-resolved photoluminescence (TRPL) spectroscopy (Fig. 3c) of IT-4F with long excited lifetime (0.79 ns), by 2.26 times of non-

fused PTIC (0.35 ns), and 1.55 times of HF-PCIC (0.51 ns). Meanwhile, FREA IT-4F has lack of protection to vinyl groups, which gives a high probability toward photooxidation in aerobic solution[30,31]. We have also tested the PL lifetime of neat films, showing 0.37 ns for IT-4F, 0.86 ns for HF-PCIC, and 1.84 ns for PTIC (Supplementary Fig. 13). PTIC in solid has nearly five times longer PL lifetime over that of IT-4F, which indicates non-fused PTIC has significantly mitigated nonradiative decay of excited state in condensed solid, wherein the vinyl group of PTIC is protected by the outward chains and tightly packed molecules. These combined factors embed PTIC with the elongated excited state and excellent photooxidative resistance. The matrix-assisted laser desorption/ionization time-of-flight (MALDI-TOF) mass spectrometry is utilized to analyze the original and photo-oxidized IT-4F films (5 h illumination, Fig. 3d). Compared with IT-4F ([M + H]$^+$, $m/z = 1500.748$), the illuminated IT-4F exhibits new signals with $m/z$ of 1515.958 and 1531.906, suggesting that one or two additional oxygen atoms are introduced, revealing the possible epoxidation of vinyl bonds for DFIC-segments[14]. As shown in Fig. 3e and Supplementary Fig. 14, Raman spectra of different films are monitored, and the vinyl group is assigned (through the comparison with IR spectra) (Supplementary Fig. 15) as 1561 cm$^{-1}$ for IT-4F and 1623 cm$^{-1}$ for HF-PCIC, both of which, along with other characteristic signals, disappeared during photodegradation. It is a sharp contrast to the stable PTIC films (1601 cm$^{-1}$) with negligible changes[32].

Together with above results from solution and film measurements, we describe the plausible photodegradation process of IT-4F in Supplementary Fig. 16. The vinyl groups of IT-4F could firstly undergo volume-conserving photoisomerization, through the concomitant rotation of C=C and C−C bonds between D and A units, which is a favorable pathway for the structural

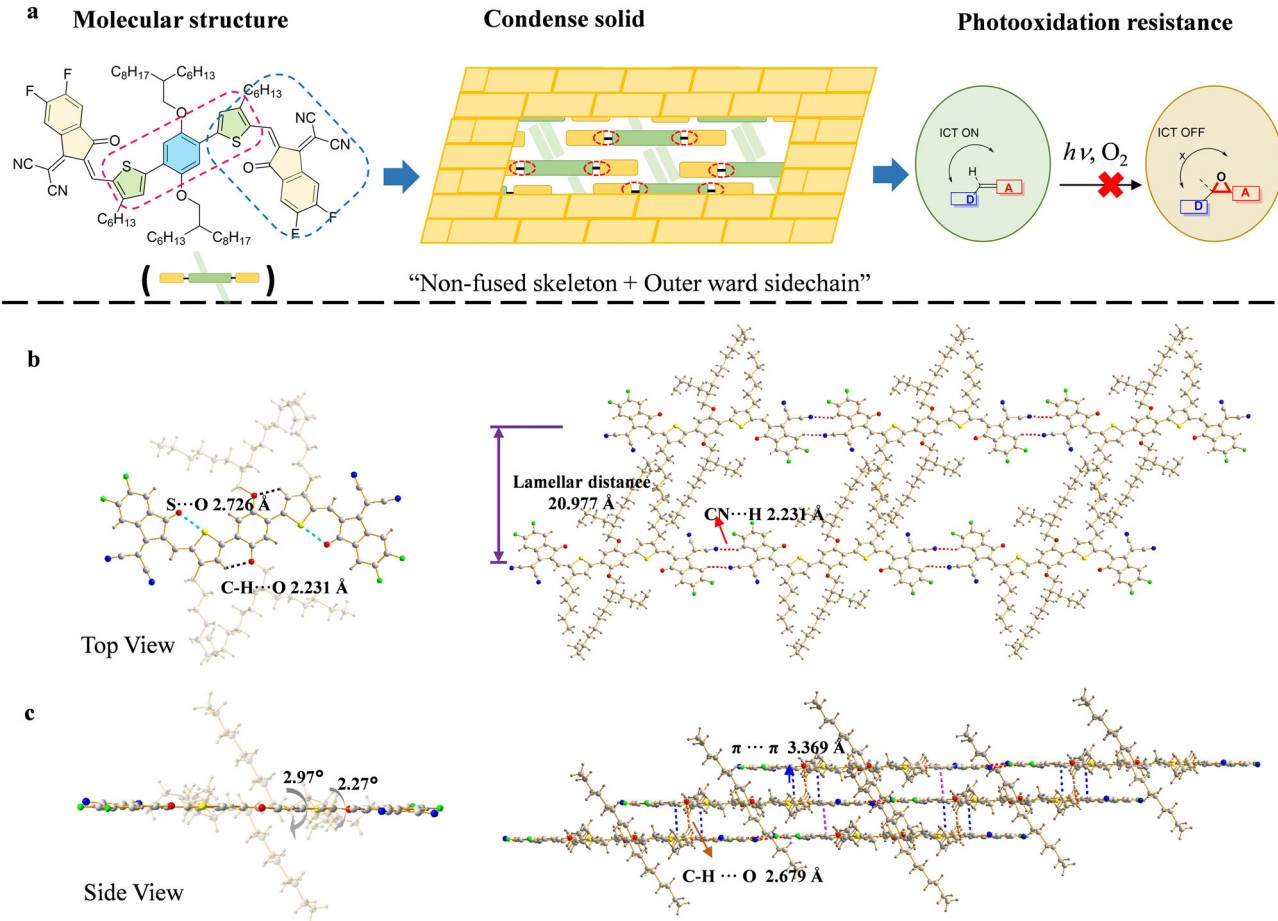

**Fig. 4 Molecular stacking of PTIC in a single crystal. a** Schematic illustration of PTIC with planar unfused structure adapts dense packing in crystals that prevents PTIC from photochemical reactions (ICT: intramolecular charge transfer). Single-crystal structure and stacking model of PTIC. **b** Top view and **c**, side view.

isomerization of the vinyl group presenting in the spatially constrained environments and/or with large end groups[33,34]. The distorted C=C bonds after photoisomerization appear to be more reactive towards photooxidation that is likely a critical step opening up the photodegradation of NFAs.

It has been observed that the distinct difference in the photostability for the studied NFAs in films versus in solution. While those electron acceptors have commonalities in optical bandgap and energy levels, PTIC exhibits exceptional photostabilities in both solution and film, over other studied examples. There are two major structural distinctions between stable PTIC and unstable IT-4F, including their backbone (non-fused versus fused) and outward thiophene side chain (with versus without). The molecular structure itself affects their solid stacking properties. Often, those crystalline films with densely packed molecules exhibit superior photostability due to the confinement of structural rearrangement in solid (Fig. 4a). To accurately analyze the solid stacking of NFAs, we performed single-crystal X-ray diffraction analysis of PTIC, which exhibits the planar geometry with C–H…O distance of 2.231 Å in the non-fused core, and with a dihedral angle of 2.97° between phenyl core and thiophene bridge. Meanwhile, the dihedral angle between thiophene and DFIC terminal is 2.27° for PTIC, which is much smaller than that of 16.1° in IT-4F crystal[35]. PTIC adapts the "brick-like" end-to-end packing (Fig. 4b) with π–π distance of 3.369 Å (blue dotted lines, Fig. 4c and Supplementary Fig. 17). Additional intermolecular interactions, including C≡N…H interaction between DFIC-segments (distance of 2.231 Å) and C–H…O interaction

(distance of 2.679 Å) facilitate the tightly packed molecules in crystals. According to grazing incidence wide-angle X-ray scattering (GIWAXS) of neat films, PTIC possesses the shortest π–π stacking distance among these studied examples (Supplementary Table 1 and Supplementary Fig. 18). The close stacking of PTIC may stem from its $Sp^3$ carbon-free planar skeleton in solid, which possesses less bulk sidechains, compared with IT-4F and HF-PCIC adapting two up- and down-point sidechains on the bridge $sp^3$-C. Therefore, the dense "brick-like" packing provides the structural confinement of PTIC in condensed solid, showing little tendency toward photoisomerization and photodegradation.

In addition to $Sp^3$ carbon-free planar backbones, we further examine the effect of terminal side chain on the photostability of NFAs, by comparing non-fused acceptors with the same backbone, PTIC (with hexyl chain) and PTICH (without hexyl chain) (Supplementary Fig. 19). Considering both exhibit good crystalline properties and long-range ordering of PTIC and PTICH from TEM and GIWAXS, it is surprising to find that the inherent photostability of PTIC is much better than that of PTICH (degradation rate of $2.30 \times 10^{-3}\,h^{-1}$, peak at 744 nm was decayed to 50% after 256 h illumination) (Supplementary Fig. 19). The possible reason behind this phenomenon should be linked to the terminal alkyl chain in the thiophene bridge, which is the only structural difference between PTIC and PTICH.

Bearing this in mind, we further develop two model molecules, TFIC and TFICH (Fig. 4a), to solely discuss the influence of alkyl group on the photostability of the vinyl group without the interference of electron-donating cores. It is not surprising to find

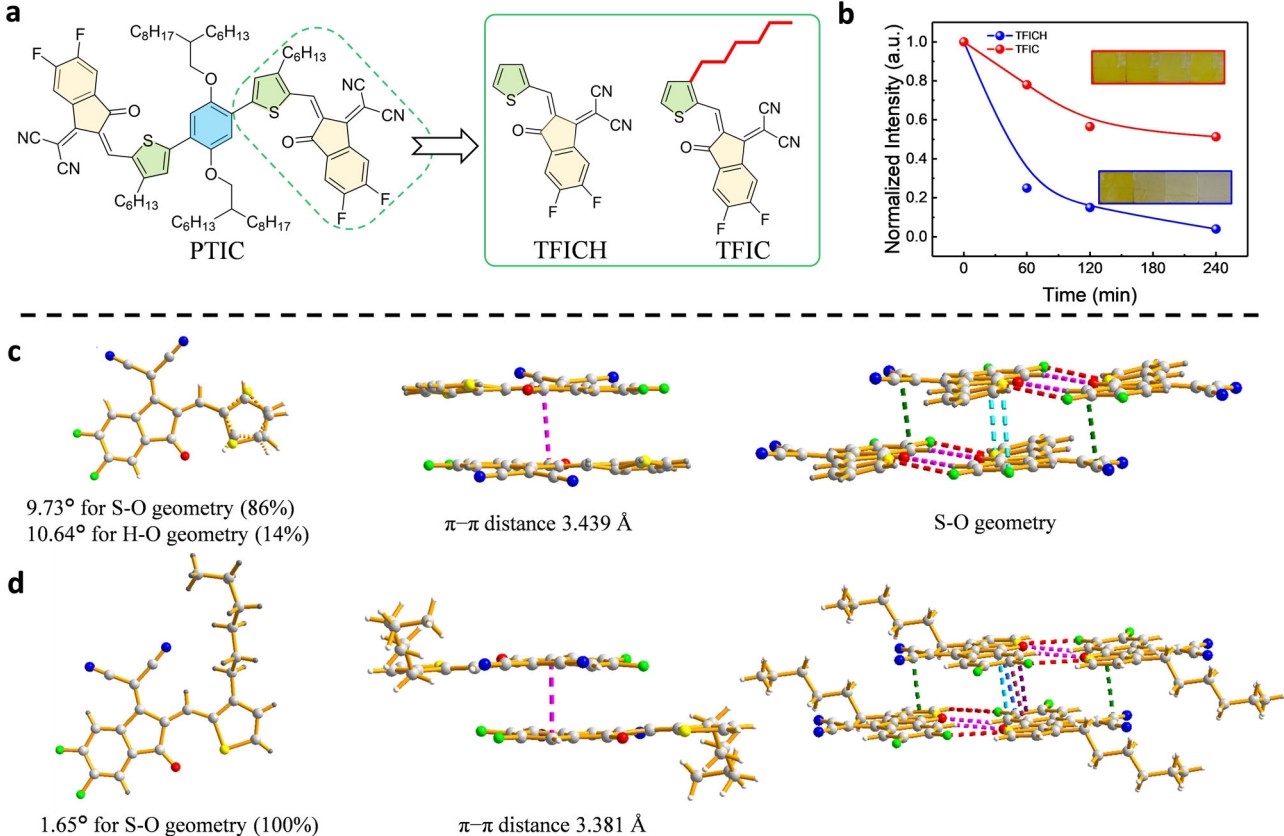

**Fig. 5 The photostability and single-crystal analysis of model compounds. a** Molecule structures of terminal models, TFICH, and TFIC from PTIC. **b** UV–vis absorption spectra and photo images of TFICH and TFIC films under continuous illumination. Single-crystal structure of **c**, TFICH and **d**, TFIC.

that the solid film of TFICH (without hexyl chain) faded to almost transparent upon 240 min of illumination, whereas TFIC (with hexyl chain) film is far more stable by remaining 50% of original absorption under the same condition (Fig. 5b). It is consistent with the decay tendency observed from PTICH and PTIC. Still, NFRA acceptors are more stable than those of model compounds, which can therefore be attributed to two main structural factors: the planar non-fused skeleton and outward hexyl chains, leading to exceptional photostabilities of A–D–A NFAs.

The single crystals of TFICH and TFIC reveal that the terminal side chain can effectively prevent the stereo-isomerization (the rotation of a single C–C bond between vinyl and thiophene[16]) of NFAs. TFIC with hexyl chains displays only one conformation with S...O geometry, whereas TFICH without hexyl chain shows stereoisomers with S...O geometry (accounting for 86% ratio) and H...O geometry (24% ratio) (Fig. 5c, d). TFICH is slightly distorted with a thiophene-DFIC dihedral angle of 9.73° for S...O geometry and 10.64° for H...O geometry, whereas TFIC shows a planar structure with a dihedral of 1.65°. Furthermore, molecular packing of TFICH demonstrates a head-to-tail stacking, and TFIC adapts a co-facial stacking of DFIC unit (The π-π distances were 3.439 Å for TFICH and 3.381 Å for TFIC, respectively).

The distorted TFICH in loosely packed solid likely results in the exocyclic double bonds more vulnerable toward photoisomerization and photooxidation. Interestingly, the photoinduced volume-conserving isomerization of vinyl group[36] is clearly observed for TFICH (R = H) in solution, transforming from Z-state (ZS) to E-state (ES) under illumination. The subsequent photooxidation of TFICH can be detected at the elongated illumination. However, the photoisomerization and photooxidation of TFIC can be largely suppressed by introducing

the terminal chain (R = C$_6$H$_{13}$) (Fig. 6a and Supplementary Figs. 20–26). Figure 6b, c shows the UV absorption changes of TFICH (R = H) and TFIC (R = C$_6$H$_{13}$) under illumination. The absorption profile and solution color of TFICH changed from yellow to deep red (due to isomerization of vinyl bond vary the intramolecular charge transfer of TFICH) in first 10 h illumination, and then appear to be dark upon the prolonged 220 h illumination (due to the generation of epoxide product, EP, along with other decomposed species). Interestingly, TFIC (R = C$_6$H$_{13}$) only exhibits little change of absorption profile and colors during the prolonged illumination.

In addition, we observed that the photoisomerism is reversible (from ES to original ZS) upon heat treatment of both TFICH and TFIC solution, which is also consistent with the temperature varied $^1$H NMR measurements (Fig. 6d, e and Supplementary Figs. 20–26). Upon rising temperature from 25 to 120 °C, the 10-h illuminated TFICH as co-mixture of ES and ZS isomers, gradually transit its ES isomer back to ZS (from $H'_a$ to $H_a$), accompanying the shift of $H_b$ (thiophene proton) from 7.714 to 7.730 ppm. When cooling down the solution from 120 °C back to 25 °C, only the ZS structure can be detected, indicating the full recovery of isomeric ES to ZS (Fig. 6d). After 10 h illumination, the ES content of TFICH is up to ~39% assigned from $^1$H NMR spectra, which is about threefolds of that in TFIC (~12%), suggesting that the hexyl chain in the thiophene bridge effectively suppresses the photoisomerization of the vinyl group. Whereas the recovery region of TFIC (at 25–80 °C range) shows less activation energy needed than that of TFICH (at 60–120 °C range) (Fig. 6e and Supplementary Figs. 27–29).

After the prolonged illumination, the epoxidation product (EP) can be detected with the characteristic chemical shift of 4.382 ppm in TFICH solution, which yields ~12% content after 50 h

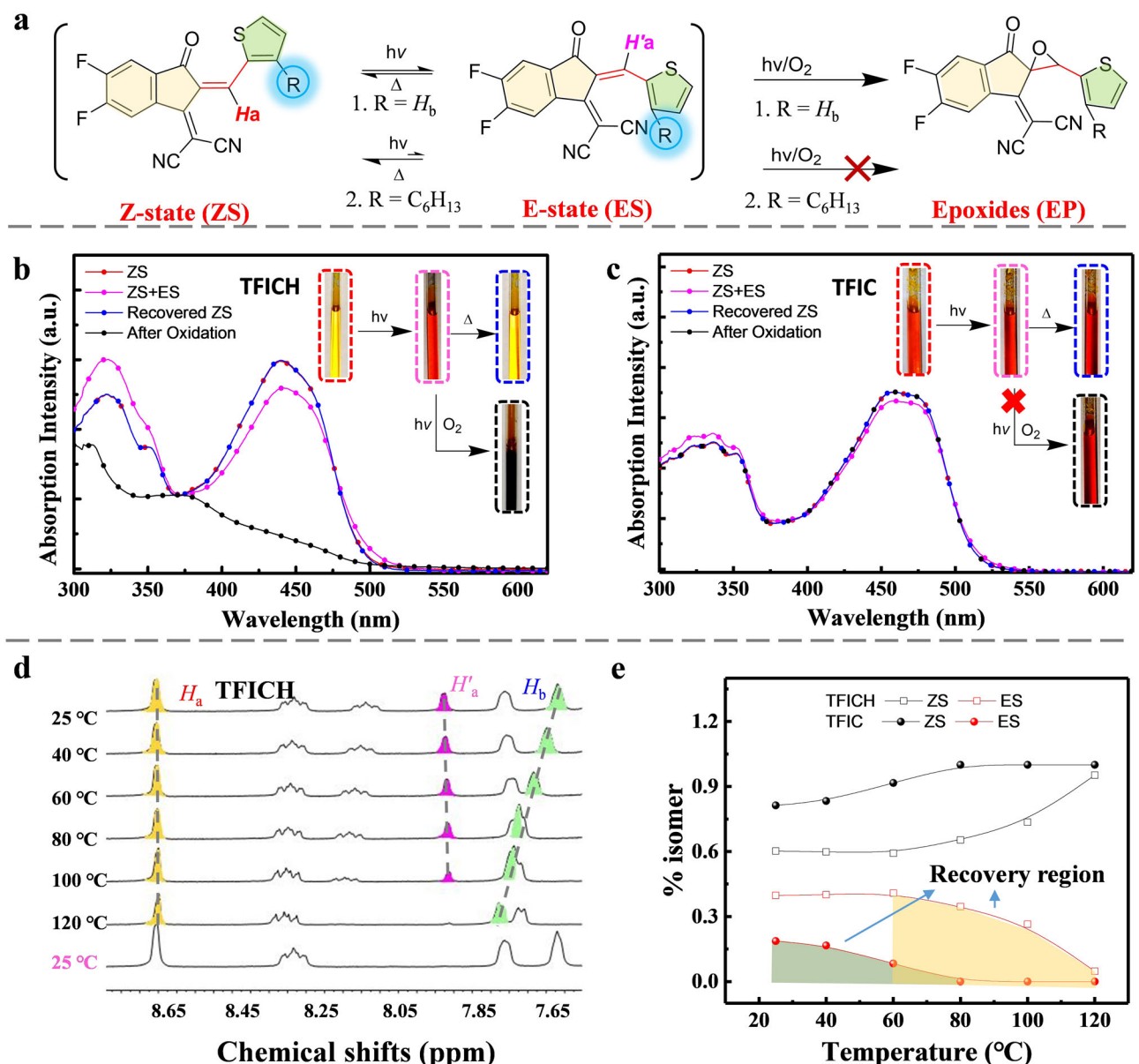

**Fig. 6 Photoisomerization of vinyl groups in model compounds. a** Photoisomerization and photooxidation of TFICH and TFIC. The photoisomerism of both **b**, TFICH and **c**, TFIC after 10 h illumination (from ZS to the co-mixture ZS + ES) that are reversible process upon heat treatment (recovered ZS), and further decomposition of TFICH after 220 h illumination (after oxidation). **d** Temperature-dependent NMR of TFICH in $d_4$-$C_6D_4Cl_2$ (heated from 25 to 120 °C, and then cooled down to 25 °C) **e**, the changes of ZS/ES isomer ratio for TFICH and TFIC upon heating treatment.

illumination. Thereafter, the content of EP-TFICH keeps increasing, accompanying the formation of more complicated signals due to the generation of other photodegraded species. Overall, this pathway of structural change involving photoisomerization and epoxidation of the vinyl group of TFICH is similar to that of IT-4F. In contrast, TFIC in solution generates little epoxidation content, and maintains mainly ZS isomers along with a very small ratio of ES isomer, after 220 h illumination. It clearly suggests that, at the molecular level, the introduction of outward-chain effectively suppresses the photoisomerization, hence the subsequent photooxidation of vinyl groups in the model compound and A–D–A NFAs.

With the above information in hands, we have further investigated how the varied NFAs influence the photostability of OSCs. Inverted OSCs are fabricated in the structure of ITO/ZnO/SAM/ active layers/MoO$_3$/Ag (Fig. 7a), wherein SAM molecules

(Supplementary Fig. 1a) are utilized to improve interfacial stability, through passivating the photocatalytic activities of ZnO of non-fullerene OSCs[37]. All those devices are fabricated under the same condition, without the usage of solvent additives. The J–V characteristics of these OSCs were measured under 100 mW cm$^{-2}$ air mass 1.5 global (AM 1.5 G) illumination (Fig. 7b, c). With the same polymer donor PBDB-TF, OSCs based on IT-4F exhibit PCE of 11% ($V_{oc} = 0.87$ V, $J_{sc} = 18.27$ mA cm$^{-2}$, FF = 0.70), which are close to those with HF-PCIC (PCE of 10.06% with $V_{oc} = 0.91$ V, $J_{sc} = 17.00$ mA cm$^{-2}$, and FF = 0.65,) and PTIC (PCE of 10.28% with $V_{oc} = 0.93$ V, $J_{sc} = 16.49$ mA cm$^{-2}$, and FF = 0.68) (Supplementary Table 2). Their photostabilities are investigated under continuous one-sun-equivalent illumination with a metal halide lamp (Fig. 7d). Slow oxygen permeation is involved for these devices tested in ambient under harsh illumination without UV filter (Supplementary Fig. 30).

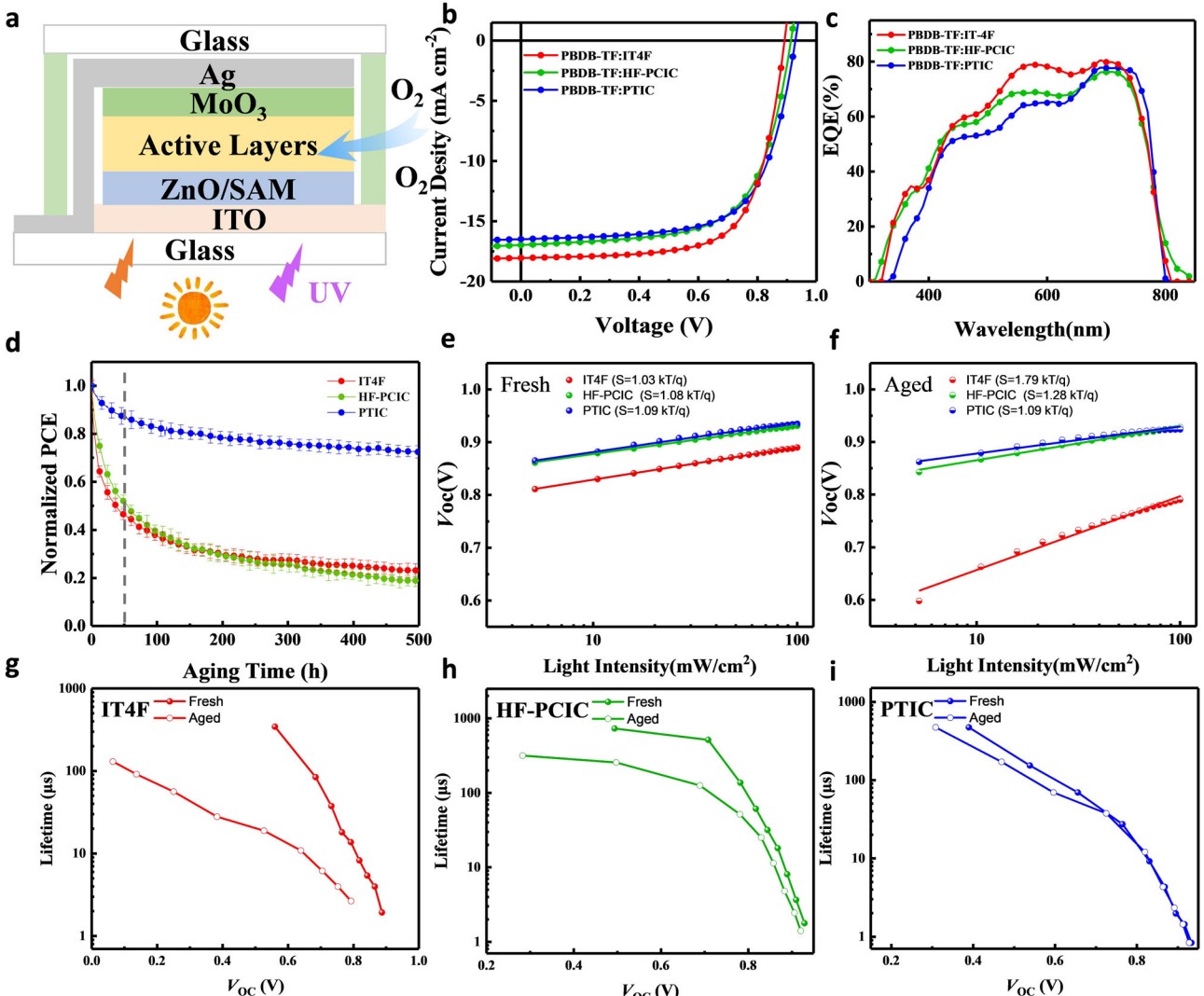

**Fig. 7 Photostability of NFOSCs. a** Schematic illustration of inverted OSCs tested in ambient without UV filter. **b** The *J–V* characteristics and **c**, EQE spectra of OSCs under AM 1.5 G illumination (100 mW cm$^{-2}$). **d** Stabilities of OSCs under continuous one-sun-equivalent illumination (the gray line indicates the first 50 h of illumination, and the error bars represent the standard deviation from four devices). $V_{oc}$ versus light intensity of **e**, fresh devices and **f**, aged devices for 50 h illumination. **g–i** Charge carrier lifetime as a function of $V_{oc}$ in fresh and aged devices for 50 h illumination based on IT-4F, HF-PCIC, and PTIC.

Surprisingly, PBDB-TF:PTIC-based device allows maintaining about 72% of its initial PCE value for 500 h illumination, which is much better than that of PBDB-TF:IT-4F and PBDB-TF:HF-PCIC-based devices (remaining only 20–30% of their initial values). Note that all measurements are subject to the freshly made devices without pre-illumination treatment, and therefore the burn-in loss of OSCs are accounted into performance decay. The fast decay of PBDB-TF:IT-4F and PBDB-TF:HF-PCIC-based devices occur at first 50 h[38], with the degradation rate of $1.08 \times 10^{-2}\,\mathrm{h}^{-1}$ and $9.66 \times 10^{-3}\,\mathrm{h}^{-1}$, respectively. In contrast, PTIC-based devices exhibit very slow decay, with the rate of $3.00 \times 10^{-5}\,\mathrm{h}^{-1}$, estimated from the first 50 h illumination. The degradation of IT-4F and HF-PCIC-based OSCs appears to be the decay of $V_{oc}$ and $J_{sc}$ parameters, ascribing to the generation of trap states due to photochemical reaction of photoactive materials with a low degree of oxygen permeation (Supplementary Fig. 30). To verify this conjecture, the *J–V* curves of fresh and aged devices are measured with the varied incident light intensity (Supplementary Fig. 31). $J_{sc}$ values as a function of light intensity is extracted to investigate the bimolecular recombination of charge carrier. The exponents α of

devices decreased obviously, for IT-4F-based devices (from 0.970 to 0.919) and HF-PCIC-based devices (from 0.969 to 0.921), whereas there is almost no influence for PTIC-based devices (from 0.952 to 0.958).

We further investigate the dependence of $V_{oc}$ on the light intensity (Fig. 7e, f), the FREA IT-4F-based devices reveal the largely increased slopes from 1.03 to 1.79 kT/q, indicating the significant loss of trap-related recombination upon photodegradation[39,40]. It is consistent with the severe $V_{oc}$ loss in photoaged OSCs. HF-PCIC-based devices also show considerable trap-assisted Shockley–Read–Hall (SRH) recombination (from 1.08 to 1.28 kT/q) after photodegradation (Supplementary Fig. 32), whereas NFEA PTIC-based OSCs have less trap-assisted recombination by remaining steady slopes of 1.09 kT/q. To further examine the charge trapping and transport behaviors of devices before and after photo illumination, charge extraction (CE) (Supplementary Fig. 33) and light-intensity-dependent transient photo-voltage (TPV) (Fig. 7g–i) are carried out[41]. CE and TPV curves of NFEA PTIC-based devices show nearly identical before and after photoaging. Whereas, FREA IT-4F and SFRA HF-PCIC-based devices show much longer charge extraction time after photoaging

than their fresh devices in the CE curves, and charge lifetime is also declined. These results indicate that abundant charge traps are generated in the photoaged devices consisting of IT-4F and HF-PCIC, which lead to the deteriorated OSC efficiencies.

The above-obtained results demonstrate that the diverse structural factors of photovoltaic molecules lead to the discrepant degradation processes under light illumination. NFRA PTIC and the derived OSCs exhibit exceptional photostabilities among the head-to-head compared with other FREA and SFRA A–D–A NFAs. We can correlate the superior photostability of NFAs and their derived OSCs to the chemical structural confinement of NFAs at the molecular level and in the solid. To investigate the applicability of this structural guidance, we extend studies into the high-performance A–D–A NFA, Y6. The identical traits of Y6 and PTIC on planar $sp^3$ carbon-free skeleton and outwards thiophene sidechains embed also good photostability of PBDB-TF:Y6-based OSCs without solvent additive (maintaining ~77% PCE of their fresh devices after 300 h illumination, Supplementary Fig. 4), as well as neat Y6 film (with 82% of the original absorption intensity under 64 h illumination in ambient, Supplementary Fig. 35).

In summary, we disclose, for the first time, that the volume-conserving photoisomerization of exocyclic vinyl groups is one critical step toward the subsequent photodegradation of a unique series of A–D–A NFAs. Among the studied NFAs with the fused, semi-fused, and non-fused backbones, non-fused PTIC displays superior photostability in solution, films, and solar cells. Non-fused PTIC-based OSCs displayed about 359 times and 322 times slower decay rates than fused IT-4F and semi-fused HF-PCIC based OSCs, respectively. It is because the structural confinement prevents the photoisomerization of NFAs at the molecular level and in condensed solid. In this regard, the structural factors, including hinder outward-chain and planar $sp^3$ carbon-free backbones play important roles to enhance the intrinsic photostabilities of these A–D–A NFAs and their derived OSCs. Even though, there are still some other complicate reasons associating with the photostabilities of non-fullerene solar cells. We believe the structure–property correlation from these unique examples can be beneficial to the community, providing new chemistry insights for designing photostable organic semiconductors and optoelectronics.

## Data availability
The authors declare that the data supporting the findings of this study are available within the paper and its Supplementary Method files.

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

## Acknowledgements
This research was funded by the National Key Research and Development Program of China (No. 2019YFA0705900), National Natural Science Foundation of China (Nos. 21722404 and 21674093), Zhejiang Natural Science Fund for Distinguished Young Scholars (LR17E030001), X.L. thanks Research Grant Council of Hong Kong (Theme-based Research Scheme No. T23-407/13-N). Z.-X. Liu thanks for the support of the 2019 Zhejiang University Academic Award for Outstanding Doctoral Candidates (2019049).

## Author contributions
C.Z.L., Z.X.L., and Z.P.Y. developed the concept and designed the experiments. C.Z.L. and H.C. supervised the project. Z.X.L. carried out the OSC fabrication and characterization. Z.P.Y. performed chemical synthesis and characterization of molecules. Z.X.L., Z.S., and C.H. characterized the sample films and solution NMR. T.K.L. and X.L. performed GIWAXS measurements. Z.C. and H.Z. performed TRPL measurements and analysis. Z.X. performed single X-ray analysis of PTIC. Z.X.L., Z.P.Y., H.C., and C.-Z.L. analyzed the results and prepared the manuscript. All authors commented on the manuscript. Z.X.L. and Z.P.Y. contributed equally to this work.

## Competing interests
The authors declare no competing interests.
