## [Peer Review File · Nature Communications]

REVIEWER COMMENTS

Reviewer #1 (Remarks to the Author):

Photostability is one of the central challenges in organic optoelectronics. This work reports an insightful investigation on how the structural factors affect the photochemical degradation of A-D-A non-fullerene acceptors (NFAs). In the combination with NMR, single crystal, spectra and devices, authors disclose that the photoisomerization of double bond plays critical role in accelerating the subsequent photooxidation of A-D-A NFAs acceptors, hence resulting in photo-degradation of acceptors, as well as the derived organic solar cells (OSCs). Due to the structural confinement from molecular design and solid dense packing, non-fused A-D-A NFAs, such as PTIC exhibit the superior photostability in solution, films and OSCs, about 359 and 322 times slower in decay rates than fused IT-4F and semi-fused HF-PCIC. Overall, this work provides insightful and new structural information on the important topic of organic materials and devices. Data presentation and analysis are well executed. Reviewer would like to recommend the acceptance of this work for the publication in Nature Communications.

Some technique comments:

- 1) The photoluminescence lifetimes of solution samples have been discussed in main text. How about the PL lifetimes of solid film? It links to the information on the photoexcited materials.
- 2) As the photoisomerization is revealed, it is better to test (or demonstrate) the point that the E-TFICH is less stable (or more reactive) than Z-TFICH.
- 3) Please check the NMR spectra of TFIC and TFICH in solution under continuous one-sun equivalent illumination in SI. Some of them may be mixed up.
- 4) Authors need to carefully check the English writing. It can be further polished.

Reviewer #2 (Remarks to the Author):

Molecular insights of exceptionally photostable electron acceptors for organic photovoltaics by Liu et al.

The manuscript by Liu et al. reports that the volume-conserving photoisomerization of exocyclic vinyl groups is one critical step towards the subsequent photodegradation of A-D-A non-fullerene acceptors (NFAs). The authors suggest that the structural confinement to inhibit such a photoisomerization of A-D-A NFAs from molecular level to macroscopic condensed solid allows eventually enhancing the photo-stabilities of molecules, as well as the corresponding NFAs. They highlight that the structural factors such as the planar non-fused skeleton and outward hexyl chains lead to exceptional photostabilities of NFAs. Based on the intrinsic photo-stabilities of A-D-A NFAs, they conclude the photostability of organic solar cells fabricated with polymer donor PBDB-TF blended with these NFAs. The observation, in particular, the photoisomerization of the vinyl groups of IT-4F, followed by photooxidation under continuous illumination is interesting, which can provide an important design rule for photostable NFAs.

The authors mentioned that the reason why PTIC and HF-PCIC are stable in solution is that their unfused backbones allow fast nonradiative relaxation of high-energy excited state, versus of those of fused structures (IT-4F) in solution. They mentioned that it is consistent with time-resolved photoluminescence (TRPL) spectroscopy (Fig. 2b) of IT-4F with long excited lifetime (0.79 ns), by 2.26 times of non-fused PTIC (0.35 ns) and 1.55 times of HF-PCIC (0.51 ns), which promotes high photooxidation probability of NFAs. It is not clear to me why unfused backbones allow fast nonradiative relaxation of high-energy excited state. More importantly why shorter excited lifetime promotes high photooxidation probability of NFAs, which is in contrast with long excited lifetime of NFAs (e.g. Y6) promoting highly efficient and stable OSC devices.

The authors have ascribed the varied photostability of different NFAs to their intrinsic structural and stacking properties, i.e. highly crystalline and dense packing producing exceptionally photostable NFAs such as PTIC. The authors concluded that the photostability of organic solar cells fabricated with polymer donor PBDB-TF and NFA blends in bulk heterojunction structures, simply

based on the intrinsic photo-stabilities of NFAs. I agree with that the intrinsic photo-stabilities of NFAs would play an important role in determining the photostability of organic solar cells. However, the structural and stacking properties, and hence their photo-stabilities of NFAs would change when they are mixed with polymers in bulk heterojunction, influencing device stability. So without detailed studies of structural and stacking properties of NFAs in bulk heterojunction blends concluding the device stability is very premature.

The authors have concluded that that abundant charge traps are generated in the photoaged devices consisting of IT-4F and HF-PCIC, which lead to the deteriorated OSC efficiencies. This is simply based on the investigation of the bimolecular recombination of charge carriers. No discussion on the cause and nature of these abundant charge traps is given.

Although it is important to identify the photoisomerization of exocyclic vinyl groups as a critical step towards the subsequent photodegradation of NFAs, I feel that the work reported here would be more suited to specialised readers such as organic synthetic chemists. I also recommend the authors would revise the paper based on the issues pointed out above.

REVIEWER REPORT(S):

Reviewer #1:

Photostability is one of the central challenges in organic optoelectronics. This work reports an insightful investigation on how the structural factors affect the photochemical degradation of A-D-A non-fullerene acceptors (NFAs). In the combination with NMR, single crystal, spectra and devices, authors disclose that the photoisomerization of double bond plays critical role in accelerating the subsequent photooxidation of A-D-A NFAs acceptors, hence resulting in photo-degradation of acceptors, as well as the derived organic solar cells (OSCs). Due to the structural confinement from molecular design and solid dense packing, non-fused A-D-A NFAs, such as PTIC exhibit the superior photostability in solution, films and OSCs, about 359 and 322 times slower in decay rates than fused IT-4F and semi-fused HF-PCIC. Overall, this work provides insightful and new structural information on the important topic of organic materials and devices. Data presentation and analysis are well executed.

Reviewer would like to recommend the acceptance of this work for the publication in Nature Communications.

Response: We thank reviewer very much for his/her positive assessments.

1. The photoluminescence lifetimes of solution samples have been discussed in main text. How about the PL lifetimes of solid film? It links to the information on the photoexcited materials.

Response: Thanks for the suggestion. We have tested the PL lifetime of solid neat films (IT-4F, HF-PCIC, PTIC). The lifetimes were 0.37 ns for IT-4F, 0.86 ns for HF-PCIC and 1.84 ns for PTIC. PTIC in solid has nearly 5 times longer PL lifetime over that of IT-4F. And PL lifetimes of neat-films (PTIC>HF-PCIC&IT-4F) are different to those in solutions (IT-4F>HF-PCIC&PTIC). These results indicate the non-fused PTIC in solid has mitigated the non-radiative decay of excited state, which can be ascribed to non-covalent interaction mediated PTIC transits into a stackable and rigid conformation in condensed solid, prohibiting the conformation changes, such as the rotation of C-C single bond. Whereas, PTIC in solution has no space confinement and

adapts the rotatable conformation with the relatively short PL lifetime, over that of fused IT-4F. This structural alternation from solution to solid would allow us designing molecular semiconductor with good balance between solution processability, and optoelectronic properties in solid states.

The corresponding results and discussion were added in the manuscript and supporting information (Page 8), as shown in Fig S13 and below:

“We have also tested the PL lifetime of neat films, showing 0.37 ns for IT-4F, 0.86 ns for HF-PCIC and 1.84 ns for PTIC (Fig. S13). PTIC in solid has nearly 5 times longer PL lifetime over that of IT-4F, which would indicate non-fused PTIC transits into a stackable and rigid conformation in condensed solid with the significantly mitigated non-radiative decay of excited state.”

Figure S13: Time-resolved photoluminescence for IT-4F, HF-PCIC and PTIC films.

2. As the photoisomerization is revealed, it is better to test (or demonstrate) the point that the E-TFICH is less stable (or more reactive) than Z-TFICH.

Response: Thanks for the suggestion. From solution measurements (UV-vis and NMR), it is observed that photo-isomerism of TFICH in solution (from ZS to ES) occurs prior to the photooxidation (after 220 hours illumination). It would serve as a good point to support that Z-form is more stable than E-form. Besides, among all the reported single crystal data, A-D-A non-fullerene acceptors all adapt Z-form of the terminal double bonds, suggesting Z-form is thermal dynamically more stable conformation than that of E-form. The isomerization of exocyclic vinyl groups occurs as volume-conserving photoisomerization in solid (Supplementary Scheme), similar to its presence in the green fluorescent protein (GFP) chromophore (J. Am. Chem. Soc. 2019, 141, 15504). We have tested the photostability of two films based on the model molecules (pure Z-TFICH and co-mixture of Z-TFICH/E-TFICH) in the supplementary Figure showing below. It found that the co-mixture film decay faster than that of pure Z-TFICH under one-sun equivalent illumination in ambient. It is also consistent with the tendency observed from solution state.

Fig. 5b: The photo-isomerism of TFICH in solution state after 10 hours illumination (from ZS to the co-mixture ZS + ES) that are reversible process upon heat treatment (Recovered ZS), and further decomposition of TFICH after 220 hours illumination (After Oxidation).

Photoisomerization of Vinyl Groups:

Supplementary Scheme: the volume-conserving photoisomerization of exocyclic vinyl groups could occur as in solid.

Supplementary Figure: Intrinsic photo stabilities of films of Z-TFICH and isomer co-mixture (Z-TFICH: E-TFICH = 1:0.61) under one-sun equivalent illumination in ambient, and the percentage of remaining absorption intensity at 425 nm.

3. Please check the NMR spectra of TFIC and TFICH in solution under continuous one-sun equivalent illumination in SI. Some of them may be mixed up.

Response: Thanks for the comments. Accordingly, we have updated Supplementary Figure 24, and corrected the sequence of the temperature-dependent ^1H NMR of TFIC-ES in Supplementary

Figure 26.

Supplementary Figure 24: The ¹H NMR of TFIC in solution under continuous one-sun equivalent illumination. Line 1-12 present spectra were recorded at different time (Fresh, 10 h, 20 h, 30 h, 40 h, 60 h, 80 h, 100 h, 130 h, 160 h, 190 h and 220 h)

Supplementary Figure 26: The temperature-dependent ¹H NMR of TFIC-ES and TFIC-IS in *d*₄-C₆D₄Cl₂ (heated from 25-120 °C and then cool down to 25 °C)

4. Authors need to carefully check the English writing. It can be further polished.

Response: Thanks for the comments. We have carefully checked the text and made updates.

Reviewer #2:

Molecular insights of exceptionally photostable electron acceptors for organic photovoltaics by Liu et al.

The manuscript by Liu et al. reports that the volume-conserving photoisomerization of exocyclic vinyl groups is one critical step towards the subsequent photodegradation of A-D-A non-fullerene acceptors (NFAs). The authors suggest that the structural confinement to inhibit such a photoisomerization of A-D-A NFAs from molecular level to macroscopic condensed solid allows eventually enhancing the photo-stabilities of molecules, as well as the corresponding NFAs. They highlight that the structural factors such as the planar non-fused skeleton and outward hexyl chains lead to exceptional photostabilities of NFAs. Based on the intrinsic photo-stabilities of A-D-A NFAs, they conclude the photostability of organic solar cells fabricated with polymer donor PBDB-TF blended with these NFAs. The observation, in particular, the photoisomerization of the vinyl groups of IT-4F, followed by photooxidation under continuous illumination is interesting, which can provide an important design rule for photostable NFAs.

Response: We thank reviewer very much for his/her positive assessments.

1. The authors mentioned that the reason why PTIC and HF-PCIC are stable in solution is that their unfused backbones allow fast nonradiative relaxation of high-energy excited state, versus of those of fused structures (IT-4F) in solution. They mentioned that it is consistent with time-resolved photoluminescence (TRPL) spectroscopy (Fig. 2b) of IT-4F with long excited lifetime (0.79 ns), by 2.26 times of non-fused PTIC (0.35 ns) and 1.55 times of HF-PCIC (0.51 ns), which promotes high photooxidation probability of NFAs. It is not clear to me why unfused backbones allow fast nonradiative relaxation of high-energy excited state. More importantly why shorter excited lifetime promotes high photooxidation probability of NFAs, which is in contrast with long excited lifetime of NFAs (e.g. Y6) promoting highly efficient and stable OSC devices.

Response: Thanks for the comments. There present clear difference for the excited non-fused acceptors in solution and in solid. (1) In solution, molecules are well dissolved. For semi-fused HF-PCIC and non-fused PTIC, there have freedom of conformation change, due to the presence of rotatable C-C bonds in backbones, which open the non-radiative decay channel of the excited molecules. The shorter life-time represent the less probability of photoreaction of excited molecule. Whereas, in solution, the fused IT-4F adapts a rigid backbone, showing PL lifetime in order of IT-4F>HF-PCIC&PTIC. Similar tendency has also presented in series of aggregation-induced emission molecules with or without the steric avoidance of non-radiative decay path ways (Chem. Rev., 2015, 115, 11718), and the bridged stilbenes (Angew. Chem. Int. Ed., 2020, 59, 10566).

(2) In the condensed solid states, the PL lifetime of films were 0.37 ns for IT-4F, 0.86 ns for HF-PCIC and 1.84 ns for PTIC (Fig. S13). PTIC in solid has nearly 5 times longer PL lifetime over that of IT-4F. And lifetimes of neat-films (PTIC>HF-PCIC&IT-4F) are different to those in solutions (IT-4F>HF-PCIC&PTIC). These results indicate the non-fused PTIC adapts a non-rotatable conformation in condensed solid (single crystal, GIWAXS and TEM), which help extending the lifetime of excited PTIC, due to prohibit the conformation change and mitigate the non-radiative decay in solid. The long-excited lifetimes of semiconducting molecules in solid are beneficial for securing long exciton diffusion length and disassociation efficiency, which are one of the necessary features for high-efficiency photovoltaic materials.

From these aspects, our observation on the solid properties of PTIC is well consistent with those of efficient NFAs, (e.g. Y6). In addition, these merits of non-fused acceptor would allow us endowing molecular semiconductors with good balance between their solution processability (soluble and stable), and optoelectronic properties in solid states.

The corresponding results and discussion were added in the manuscript and supporting information (Page 8), as shown in Fig S13.

“We have also tested the PL lifetime of neat films, showing 0.37 ns for IT-4F, 0.86 ns for HF-PCIC and 1.84 ns for PTIC (Fig. S13). PTIC in solid has nearly 5 times longer PL lifetime over that of IT-4F, which would indicate non-fused PTIC transits into a stackable and rigid

conformation in condensed sloid in solid with the significantly mitigated non-radiative decay of excited state.”

Supplementary Figure 13: Time-resolved photoluminescence for IT-4F, HF-PCIC and PTIC in films.

2. The authors have ascribed the varied photostability of different NFAs to their intrinsic structural and stacking properties, *i.e.* highly crystalline and dense packing producing exceptionally photostable NFAs such as PTIC. The authors concluded that the photostability of organic solar cells fabricated with polymer donor PBDB-TF and NFA blends in bulk heterojunction structures, simply based on the intrinsic photo-stabilities of NFAs. I agree with that the intrinsic photo-stabilities of NFAs would play an important role in determining the photostability of organic solar cells. However, the structural and stacking properties, and hence their photo-stabilities of NFAs would change when they are mixed with polymers in bulk heterojunction, influencing device stability. So without detailed studies of structural and stacking properties of NFAs in bulk heterojunction blends concluding the device stability is very premature.

Response: Thanks for the comment and suggestion. We agree with reviewers that the structural stacking, and morphological properties (as inherent factors) of photovoltaic blends play

synergistic roles in governing the photo-stabilities of NFA based solar cells. In our studies, the intrinsic photostability on the chemical structure of acceptors appears to be primary factor governing the operational stability of photovoltaic devices, which, of course, has close consequence to our testing condition with a metal halide lamp without UV-filtration in ISOS-L-3 testing setup. Form our experimental observation of these examples, the photoisomerization and subsequent photoreaction of acceptors occurs fast enough, which is prior to the stage allows discussing the stacking and morphological properties of blends.

As shown in Fig. S3 and S4, the intrinsic photostability of polymer and blend films (unencapsulated and encapsulated) have been monitored under the one-sun equivalent illumination in ambient. For the unencapsulated samples (Fig. S3), the absorption peaks of PBDB-TF, PBDB-TF:IT-4F and PBDB-TF:HF-PCIC films have all disappeared under 128 hours illumination, suggesting the photo-degradation of both polymer and acceptors. To our surprise, clear peaks representing PTIC absorption remain in PBDB-TF:PTIC films, despite of the disappear of polymer absorption, which suggest the decent photo-stabilities of PTIC.

We have conducted same measurements with the encapsulated films (same as the device encapsulation condition in this work, Fig. S4). It shows that IT-4F degrades after 100 hours illumination, likely photoisomerization and photodegradation, involving photooxidation with oxygen residue, dimerization and decomposition etc. And the similar trend appears in PBDB-TF:IT-4F blend. PTIC film and PBDB-TF:PTIC blend maintain good stability under such testing condition. To obtained more information, photoluminescence of neat films was measured (Supplementary Figure 5). IT-4F film show a completely different emission profile with the significant blue-shift peak at 553 nm after aging, while the original emission peaks at 792 nm. PTIC films remain identical PL profiles before and after aging. It, again, suggests that PTIC possess superior photostability.

From these results, we believe that the intrinsic photostability of NFAs (photoisomerization and photodegradation) is the primary factor to determine the photostability of devices under our measuring conditions. Under such condition, there makes less necessary to further probe the detailed information on the stacking and morphological properties of NFAs in blends.

The corresponding results and discussion were added in the manuscript and supporting

information (Line 10-19, Page 6), as shown in Fig. S4 and Fig. S5.

“Similar decay tendency has been observed in the encapsulated films (Fig. S4), which, again, show PTIC film and PBDB-TF:PTIC blend possess superior photostability. To obtain more information, photoluminescence (PL) of neat films was measured (Fig. S5). PTIC films remain identical PL profiles before and after aging. Whereas, IT4F film show a completely different emission profile with significant blue-shift peaks after aging.”

Supplementary Figure 3: UV-vis absorption spectra of thin films of polymer and blend films under continuous one-sun equivalent illumination.

Supplementary Figure 4: UV - vis absorption spectra and photo images of encapsulated films (100 °C annealing for 10 minutes and then encapsulated in glovebox) under one-sun equivalent illumination in ambient.

Supplementary Figure 5: Photoluminescence of encapsulated films before and after 100 hours one-sun equivalent illumination in ambient.

3. The authors have concluded that that abundant charge traps are generated in the photoaged devices consisting of IT-4F and HF-PCIC, which lead to the deteriorated OSC efficiencies.

This is simply based on the investigation of the bimolecular recombination of charge carriers. No discussion on the cause and nature of these abundant charge traps is given.

Response: Thanks for the suggestion. The decay of device performance has direct links to the structural degradation of active layers, especially associated with photoisomerization and photodegradation of acceptors. As shown in the above supplementary figures, we found the photo reaction of acceptors happened at early stage, even before allowing us considering the stacking and morphological change of blends. Photoluminescence (PL) measurements indicate that the degraded IT4F films possess significantly blue-shift emission peaks to 553 nm (original peak at 792 nm). It suggested that the degraded species have large shrinkage of π -conjugated aromatic systems, which would associate with the deterioration of optoelectronic and charge transport properties of photovoltaic materials, hence impacting on the device performance.

The corresponding results and discussion were added in the manuscript and supporting information (Page 6).

“It would suggest that the degraded species have large shrinkage of π -conjugated aromatic systems, associating with the deterioration of optoelectronic and charge transport properties of photovoltaic materials, hence impacting on the device performance.”

4. Although it is important to identify the photoisomerization of exocyclic vinyl groups as a critical step towards the subsequent photodegradation of NFAs, I feel that the work reported here would be more suited to specialised readers such as organic synthetic chemists. I also recommend the authors would revise the paper based on the issues pointed out above.

Response: Thanks for the suggestions. Even after large number of new acceptors being developed, few works are focused on the photostability of material themselves, in particular, there lacks of knowledge that how to access A-D-A NFAs with excellent resistance towards photochemical degradation. We believe these efforts should be highly valuable for community to further develop stable and efficient organic semiconductors and the related optoelectronics. As reviewer 1 also noted, photostability is one of the central challenges in organic optoelectronics. And recommend that this work provides insightful and new structural information on the important topic of organic materials, and is in favor of understanding the stable devices. It is true the intrinsic photostability of semiconducting molecules is the most critical factor in

governing the photo-stabilities of solar cells. Herein, we disclose, for the first time, that volume-conserving photoisomerization of A-D-A NFAs acts as a surrogate towards the subsequent photodegradation, in particular, epoxidation of vinyl double bonds of acceptors, hence resulting in degradation of the derived OSCs. Along with a series of in-depth investigations, we also identify that the structural confinement to inhibit photoisomerization of A-D-A NFAs from molecular level to macroscopic condensed solid allows eventually enhancing the photo-stabilities of molecules, as well as the corresponding non-fullerene OSCs. We believe these valuable insights should be beneficial to the broad community of readership, which can direct new design of stable and efficient organic semiconductors and solar cells. Therefore, it is well justified to Nature Communications.

Reviewers' comments:

Reviewer #1 (Remarks to the Author):

The authors have addressed the issues in the previous round of review, and the paper can be accepted now.

Reviewer #2 (Remarks to the Author):

The manuscript reports the photoisomerization of exocyclic vinyl groups as a critical step towards the subsequent photodegradation of NFAs. The work reported here is interesting. The authors have revised the manuscript responding to the questions raised previously. However, the main questions are still not properly addressed and some of them are listed below. It still requires further explanation and clarification. I do not recommend publishing this manuscript as it is in this journal. The authors should revise the manuscript based on the issues pointed out below before considering resubmission even to another journal.

1. PL lifetime: The authors stated IT-4F with long-excited lifetime (0.79 ns), by 2.26 times of non-fused PTIC (0.35 ns) and 1.55 times of HF-PCIC (0.51 ns) promotes high photooxidation probability of NFAs leading to photoreaction. Then they measured the longest lifetime in PTIC (and 1.84 ns compared to 0.37 ns for IT-4F and 0.86 ns for HF-PCIC) and explained that the long-excited state lifetime is beneficial for high-efficiency and high stability photovoltaics. This sounds very contradictory. Why does long-lived excited state in film not lead to more degradation as found in solution?

2. Photoisomerization: The authors stated that the distorted C=C bonds after photoisomerization appears to be more reactive towards photooxidation that is likely a critical step opening up the photo degradation of NFAs. However, they have not explained why this is the case for their materials. Why does photoisomerization increase the propensity for photooxidation? I feel this is a key observation in the paper but it hasn't been explained.

3. Molecular structure: The authors stated that the planar non-fused skeleton and outward hexyl chains lead to exceptional photostability of A-D-A NFAs. I am not convinced by the fused vs unfused ring explanation for better photostability, except that the unfused structure has less "pointy out" side groups, allowing better packing. The authors cited that the fused ring acceptors Y6, fits well with their explanation. However, the stability of Y6 is reported to be due to the restricted rotation induced by the side chains.

4. Raman spectra: The authors showed Raman spectra of different films and indicated the vinyl group (assigned to 1561 cm⁻¹ for IT-4F and 1623 cm⁻¹ for HF-PCIC) along with other characteristic signals, disappeared during photodegradation. However, the IT-4F Raman spectra are not clear at all, as there seems to be no clear changes in relative peak intensities, instead all peaks decrease their intensities.

5. BHJ morphology: The authors provided more optical data to show photo-stabilities of NFAs. However, it still does not provide the information about other critical factors such as BHJ morphology which is known to strongly influence device stability. The detailed studies of structural and stacking properties of NFAs in BHJ blends are needed to exclude them as main contributor for the device stability.

6. Devices: The authors discussed about photooxidation but then they looked at encapsulated devices (figure 6).

7. There are many English errors throughout the manuscript. The authors should check carefully.

REVIEWER REPORT(S):

Reviewer #1 (Remarks to the Author):

The authors have addressed the issues in the previous round of review, and the paper can be accepted now.

Response: Thank the reviewer very much for his/her positive assessments.

Reviewer #2 (Remarks to the Author):

The manuscript reports the photoisomerization of exocyclic vinyl groups as a critical step towards the subsequent photodegradation of NFAs. The work reported here is interesting. The authors have revised the manuscript responding to the questions raised previously. However, the main questions are still not properly addressed and some of them are listed below. It still requires further explanation and clarification. I do not recommend publishing this manuscript as it is in this journal. The authors should revise the manuscript based on the issues pointed out below before considering resubmission even to another journal.

Response: Thank the reviewer very much for his/her positive assessments. This study focuses on the photooxidation degradation of non-fullerene molecules under aerobic condition. To the best of our knowledge, few works touch the photochemical stability of non-fullerene materials themselves that appears to be one of primary factors to govern the overall stability of devices. In this work, we have revealed that the structural confinement to inhibit photoisomerization of A-D-A NFAs from molecular level to macroscopic condensed solid allows eventually enhancing the photochemical stabilities of NFAs, as well as the corresponding OSCs.

We have provided clear experimental evidence, along with a series of in-depth investigation, to reveal that two structural factors for securing photochemical stability of NFAs under aerobic condition: 1) to suppress the photoisomerization of vinyl groups by installing outward-chain in A-D-A NFAs; 2) to promote dense packing of molecules with planar sp^3 carbon-free backbones. These insights would allow inspiring new design of stable and efficient organic photovoltaic materials. We believe, as also positively supported by other reviewer, this work is well justified for publication in a prominent journal, Nature Communications.

1. PL lifetime: The authors stated IT-4F with long-excited lifetime (0.79 ns), by 2.26 times of non-fused PTIC (0.35 ns) and 1.55 times of HF-PCIC (0.51 ns) promotes high photooxidation probability of NFAs leading to photoreaction. Then they measured the longest lifetime in PTIC (and 1.84 ns compared to 0.37 ns for IT-4F and 0.86 ns for HF-PCIC) and explained that the long-excited state lifetime is beneficial for high-efficiency and high stability photovoltaics. This sounds very contradictory. Why does long-lived excited state in film not lead to more degradation as found in solution?

Response: Thanks for the question. This comment is previously raised by Reviewer 1 that was well addressed. In this work, we have experimentally disclosed the photochemical degradation of

NFAs is linked to the structural factors of molecules, and come up solutions: 1) to suppress the photoisomerization of vinyl groups by installing outward-chain in A-D-A NFAs (Scheme 1d in the main text); 2) to promote dense packing of molecules with planar sp^3 carbon-free backbones (Scheme 1e in the main text).

As for reviewer question, why does long-lived excited state in film not lead to more degradation as found in solution? PL lifetime has correlation, but not the sole factor to influence the photochemical reaction of molecules. Non-fused PTIC appears with both good photochemical stability and relatively long PL lifetime in solid. They are not contradictory factors. It is because non-fused acceptors, as discussed in main text, are mediated by non-covalent interaction that adapt a rotatable conformation in solution and transit into planar conformation with dense stacking in solid. Therefore, there appears difference for the PL lifetime of non-fused acceptors in solution and in solid.

(1) In solution, molecules are dissolved, and surrounded by solvent. For semi-fused HF-PCIC and non-fused PTIC, there have rotatable single-bond in backbone, which are incline to quench the excited state via the non-radiative decay channel, for instance molecular vibration and motion. Whereas, fused molecule itself with rigid and chemical-bond locked backbone appears with long-excited life time in solution. Meanwhile, IT-4F has lack of protection to vinyl groups, which gives high probability towards photoisomerization and photooxidation, as revealed by NMR and TRPL measurements.

(2) In condensed solid, non-fused PTICs were tightly stacked with planar conformation in solid (as revealed by crystal, XRD and TEM), wherein the vinyl groups are well protected by outward-chain and packed molecules. The molecular motion is also constrained to mitigate non-radiative decay channel. Therefore, PL lifetimes of neat-films for PTIC is largely elongated to that in solutions. These combined factors embed PTIC with the elongated and photooxidation-resistive excited state that is beneficial for the stable and efficient photo-to-electron conversion of solar cells.

Overall, non-fused acceptors, representing one of rare existing samples, exhibit appearing characteristics of both good photochemical tolerance and long excited lifetime in solid. In addition, they can be prepared through the extremely simple chemistry. These pleasant properties are not contradictory factors, just needs more understandings and explorations. We believe the insights reported herein are beneficial for community to further develop stable and efficient organic photovoltaic materials.

Regarding to reviewer comments, description has been added into main text:

It is consistent with time-resolved photoluminescence (TRPL) spectroscopy (Fig. 2b) of IT-4F with long excited lifetime (0.79 ns), by 2.26 times of non-fused PTIC (0.35 ns) and 1.55 times of HF-PCIC (0.51 ns). Meanwhile, FREA IT-4F has lack of protection to vinyl groups, which gives high probability towards photooxidation in aerobic solution.

PTIC in solid has nearly 5 times longer PL lifetime over that of IT-4F, which indicate non-fused PTIC has significantly mitigated non-radiative decay of excited state in condensed solid, wherein the vinyl group of PTIC is well protected by the outward-chains and tightly packed molecules. These combined factors embed PTIC with the elongated excited state and excellent photooxidative resistance.

2. Photoisomerization: The authors stated that the distorted C=C bonds after photoisomerization

appears to be more reactive towards photooxidation that is likely a critical step opening up the photo degradation of NFAs. However, they have not explained why this is the case for their materials. Why does photoisomerization increase the propensity for photooxidation? I feel this is a key observation in the paper but it hasn't been explained. The outcomes of this work are far more than the above description.

Response: Thanks for bringing this again. This comment is also asked by Reviewer 1 that was previously well addressed. From UV-vis and NMR measurements, it clearly shows that photo-isomerism of TFICH from ZS to ES occurs prior to the photooxidation (after 220 hours illumination). This experimental evidence serves as a solid foundation to support that Z-form is more stable than E-form. Besides, among all the reported single crystal data, A-D-A non-fullerene acceptors all adapt Z-form of the terminal double bonds, suggesting Z-form is thermal dynamically more stable conformation than that of E-form. The isomerization of exocyclic vinyl groups occurs as volume-conserving photoisomerization in solid (Supplementary Scheme), which is similar to its presence in the green fluorescent protein (GFP) chromophore (J. Am. Chem. Soc. 2019, 141, 15504). We have tested the photostability of two films based on the model molecules (pure Z-TFICH and co-mixture of Z-TFICH/E-TFICH) in the supplementary Figure showing below. It is found that the co-mixture film decay faster than that of pure Z-TFICH under one-sun equivalent illumination in ambient. It is also consistent with the tendency observed from solution state. Once we introduced alkyl group, TFIC only exhibits little change of absorption profile and colors during the prolonged illumination and little epoxidation content can be observed even tracking by ^1H NMR spectra. It clearly suggests that, at molecular level, suppression of the photoisomerization can be an effective method to suppress the photooxidation of A-D-A NFAs.

Fig. 5b: The photo-isomerism of TFICH in solution state after 10 hours illumination (from ZS to the co-mixture ZS + ES) that are reversible process upon heat treatment (Recovered ZS), and further decomposition of TFICH after 220 hours illumination (After Oxidation).

Photoisomerization of Vinyl Groups:

Supplementary Scheme: the volume-conserving photoisomerization of exocyclic vinyl groups could occur as in solid.

Supplementary Figure: Intrinsic photo stabilities of films of Z-TFICH and isomer co-mixture (Z-TFICH: E-TFICH = 1:0.61) under one-sun equivalent illumination in ambient, and the percentage of remaining absorption intensity at 425 nm.

3. Molecular structure: The authors stated that the planar non-fused skeleton and outward hexyl chains lead to exceptional photostability of A-D-A NFAs. I am not convinced by the fused vs unfused ring explanation for better photostability, except that the unfused structure has less “pointy out” side groups, allowing better packing. The authors cited that the fused ring acceptors Y6, fits well with their explanation. However, the stability of Y6 is reported to be due to the restricted rotation induced by the side chains.

Response: Thanks. We think this is not the correct interpretation of our work. What we have discussed in main text is there are two primary factors for achieving the photochemical stability of A-D-A NFAs: 1) to suppress the photoisomerization of vinyl groups by installing outward-chain in A-D-A NFAs (Scheme 1d in the main text); 2) to promote dense packing of molecules with planar sp^3 carbon-free backbones (Scheme 1e in the main text). From molecular level to macroscopic condensed solid, PTIC and Y6 (as we have discussed, along with others studies) fit well with this description.

4. Raman spectra: The authors showed Raman spectra of different films and indicated the vinyl group (assigned to 1561 cm^{-1} for IT-4F and 1623 cm^{-1} for HF-PCIC) along with other characteristic signals, disappeared during photodegradation. However, the IT-4F Raman spectra are not clear at all, as there seems to be no clear changes in relative peak intensities, instead all peaks decrease their intensities.

Response: Thanks for the comments. The Raman signals are sensitive to film properties. IT-4F spectra appears with good signal-to-noise ratio, as can be assigned to 1561 cm^{-1} for vinyl bond at initial stage, which is disappeared due to fast degradation under illumination in ambient. We have added a supplementary figure including Infrared spectrum and Raman spectrum. These complementary signals are clear enough to assign molecular structures.

Supplementary figure: Raman spectrum (blue curves) and IR spectrum (red curves) of NFAs, The peaks of marked by green line represents the signals of exocyclic double bond (IT-4F 1561 cm^{-1} , HF-PCIC 1561 cm^{-1} , PTIC 1601 cm^{-1}).

5. BHJ morphology: The authors provided more optical data to show photo-stabilities of NFAs. However, it still does not provide the information about other critical factors such as BHJ morphology which is known to strongly influence device stability. The detailed studies of structural and stacking properties of NFAs in BHJ blends are needed to exclude them as main contributor for the device stability.

Response: Thanks for the comments. This study focused on the photooxidation degradation of molecules. We have revealed, along with a series of in-depth investigations, that the structural confinement to inhibit photoisomerization of A-D-A NFAs from molecular level to macroscopic condensed solid allows eventually enhancing the photochemical stabilities of molecules. We agree with reviewer that the BHJ morphological properties are important factors to influence performance-stabilities of solar cells, which, however, should build on the photostable materials. In our experiment setting, the photochemical stability of acceptors is discussed under aerobic condition with a metal halide lamp illumination (without UV-filtration). Even for the simple encapsulated prototype OSC devices, low degree of oxygen permeation can not be avoided. And we showed that the photochemical reaction of encapsulated film examples occurs fast (Supplementary Figure). On this basis, we think there gave no point to further spend months for studying their morphological information. Hope reviewer also agree, to require detailed studies on morphological information is somehow beyond the scope of this work.

Description has been added into main text:

The degradation of IT-4F and HF-PCIC based OSCs appears to be the decay of V_{oc} and J_{sc} parameters, ascribing to the generation of trap states due to photochemical reaction of photoactive materials with low degree of oxygen permeation (Fig. S30).

Supplementary Figure: UV-vis absorption spectra and photo images of encapsulated films (100 °C annealing for 10 minutes and then encapsulated in glovebox with epoxy sealant) under one-sun equivalent illumination in ambient.

6. Devices: The authors discussed about photooxidation but then they looked at encapsulated devices (figure 6).

Response: Thanks for the comments. We believe low degree of oxygen permeation is involved for these OSC devices tested in ambient without UV-filtration. Under same encapsulation, we have shown that the photo degradation of film examples occurs similarly, yet slowly, to those without encapsulation (Above supplementary figure). Oxygen is the primary killer for the photoexcited organic semiconductors. Exceptionally, this study has revealed non-fused acceptors, representing one of rare existing samples, exhibit appearing characteristics of both good photochemical tolerance and photovoltaic performance. We mainly focused on the investigation of photooxidation degradation of molecules under aerobic condition, and disclosed that the structural confinement to inhibit photoisomerization of A-D-A NFAs from molecular level to

macroscopic condensed solid allows eventually enhancing the photochemical stabilities of molecules. So far, few works touch this important factor. We believe this work brings new insights and understandings towards the future development of organic photovoltaic materials and solar cells.

Description has been added into main text:

Slow oxygen permeation is involved for these devices tested in ambient under harsh illumination without UV- filter (Fig. S30).

Updated Figure 6

Fig. 6 | Photostability of NFOSCs. *a*, Schematic illustration of inverted OSCs tested in ambient without UV-filtration; *b*, The J - V characteristics and *c*, EQE spectra of OSCs under AM 1.5 G illumination (100 mW cm^{-2}). *d*, Stabilities of OSCs under continuous one-sun equivalent illumination (The gray line indicates the first 50 hours of illumination, and the error bars represent the standard deviation from four devices). V_{oc} versus light intensity of *e*, fresh devices and *f*, aged devices for 50 hours illumination; *g-i*, Charge carrier lifetime as a function of V_{oc} in fresh and aged devices for 50 hours illumination based on IT-4F, HF-PCIC and PTIC.

7. There are many English errors throughout the manuscript. The authors should check carefully.

Response: Thanks for comments. We have carefully checked manuscript and made updates.

REVIEWERS' COMMENTS

Reviewer #3 (Remarks to the Author):

In the past few years, the field of organic solar cells achieved much progress due to the development of non-fullerene acceptors. As the efficiencies are quite high, dealing with the stability problems becomes more urgent. This study discussed the chemical stability of the non-fullerene acceptors and provides some insightful results. I carefully checked the comments from reviewer #2 and the corresponding response of the authors. I think the characterizations were well performed and the manuscript was well prepared. I support the publication of this manuscript as I think the results are meaningful and timely although there are debates on some details. The followings list some small questions. 1) About the photostability test, more details about the conditions should be provided, such as the spectra range of the lamp, the film thickness. Does the humidity affect the stability? 2) I noted that the polymer PBDB-TF also shows quick photodegradation. The authors should put more discussion on why? What are the differences when compared to the non-fullerenes, fused, or non-fused? 3) After the illumination treatment, do the degraded fragments volatilized? The optical density is nearly zero. 4) About the conclusion, I think it's better to point out there may be other reasons that affecting the chemical stability of these materials as the studied systems are quite limited when compared to the boom of OSC materials. 5) I think the authors should update the references as the field is growing very fast. It's better to conclude the latest progress reporting record efficiencies, such as 17-18% efficiencies obtained using non-fullerene acceptors like bo-4cl, eC9.

REVIEWER REPORT(S):

Reviewer #3 (Remarks to the Author):

In the past few years, the field of organic solar cells achieved much progress due to the development of non-fullerene acceptors. As the efficiencies are quite high, dealing with the stability problems becomes more urgent. This study discussed the chemical stability of the non-fullerene acceptors and provides some insightful results. I carefully checked the comments from reviewer #2 and the corresponding response of the authors. I think the characterizations were well performed and the manuscript was well prepared. I support the publication of this manuscript as I think the results are meaningful and timely although there are debates on some details. The followings list some small questions.

Response: Thank the reviewer very much for his/her positive assessments.

1. About the photostability test, more details about the conditions should be provided, such as the spectra range of the lamp, the film thickness. Does the humidity affect the stability?

Response: Thanks for the comments. We have incorporated the spectra of metal halide lamp in the supporting information (Supplementary Figure 2). In the meanwhile, sample films have been measured by depth profiles that show similar thickness. Values have been added into SI (IT-4F film of ~81 nm, HF-PCIC of ~ 83 nm and PTIC ~ 82 nm) and the blend films (PM6: IT-4F of ~108 nm), PM6: HF-PCIC of ~112 nm and PM6: PTIC of ~109 nm), respectively.

We believe the humidity plays roles in affecting the photostability of organic semiconductors, which, however, is not the main cause of degradation in this study. Photo-oxidation is the main factor that leads to the fast decomposition of samples. The experiments have been conducted in clean room, wherein the humidity is under relatively well control. Through the comparison of the bare films and encapsulated films with epoxy sealants, we have observed the similar trend of degradation kinetics. Oxygen has faster permeation rate than that of water, which would support that photo-oxidation plays main roles to degrade samples in these experiments.

Regarding to reviewer comments, additional description has been added into supporting information:

Supplementary Figure 3: UV-vis absorption spectra of neat polymer films and blend films with the thickness of ~ 110 nm (PBDB-TF of 92 nm, PBDB-TF: IT-4F of 108 nm, PBDB-TF: HF-PCIC of 112 nm and PBDB-TF: PTIC of 109 nm) and the thickness of ~ 80 nm for neat acceptor films (IT-4F of 81nm, HF-PCIC of 83 nm and PTIC of 82 nm) under continuous one-sun equivalent illumination..

Supplementary Figure 2: Light spectrum of a metal halide lamp (PHILIPS MSR 1200HR) in ISOS-L-3 type photostability testing setup (Infinity PV).

2. I noted that the polymer PBDB-TF also shows quick photodegradation. The authors should put more discussion on why? What are the differences when compared to the non-fullerenes, fused, or non-fused?

Response: Thanks for the suggestions. It is worth to note that the light spectra we employed in this study contains high portion of ultraviolet photons (250-380 nm), which is the common source of instability of organic materials that associate chemical bond dissociation when exposing to high energy (i.e. UV) radiation (Adv. Energy Mater. 2011, 1, 491; Adv. Energy Mater. 2019, 9, 1902124). A number of articles have investigated the photostability of conjugated polymers. Taking P3HT as example, the photochemical degradation involves a chain radical oxidation process (Thin Solid Films, 2010, 518, 7113). We think the conjugated polymer PM6 as electron donor would follow the similar degradation pathway. Whereas, the detailed studies on polymer donors are somehow, beyond the scope of this work.

It is true that, combining the harsh condition of high energy UV with open air, conventional wisdoms told us organic materials would not stand long until decomposition. Exceptionally, in this work, we reveal that non-fused PTIC, representing one of rare existing samples, exhibits excellent characteristics of both superior photochemical tolerance and photovoltaic performance. To gain understanding on this unusual, yet important properties, we identify that the structural confinement to inhibit photoisomerization of A-D-A NFAs from molecular level to macroscopic condensed solid allows eventually enhancing the photochemical stabilities of molecules (Fig. 3). Regarding to structure difference among fused and non-fused, we found 1) outward-chains on thiophene bridge help suppressing the isomerization of vinyl groups in A-D-A NFAs; 2) Sp³ carbon-free backbones promote dense packing of molecules, and mitigated the chain radical oxidation. Hope the structural-properties correlation from these unique examples can be beneficial to the community on developing superior organic photovoltaic materials. According to International Electrotechnical Commission (IEC) standard for terrestrial photovoltaic modules, solar cells need to verified over long period against ultraviolet (UV) irradiance under thermal stress, wherein the photochemical stabilities of molecules are the important inherent factors to determine the overall stability of photovoltaics.

Regarding to reviewer comments, we have added description into main text:

Original: The photostability of PTIC remains distinct as well in the bulk heterojunction blend with polymer donor, PBDB-TF, wherein PTIC maintains absorption characteristics well while the fast decay of polymer (Fig. S3). Similar decay tendency has been observed in the encapsulated films (Fig. S4), which, again, show PTIC film and PBDB-TF:PTIC blend possess superior photostability.

Revised: The photostability of PTIC remains distinct as well in the bulk heterojunction blend with polymer donor, PBDB-TF, wherein PTIC maintains absorption characteristics well while the fast decay of polymer (Fig. S3). Similar decay tendency has been observed in the encapsulated films (Fig. S4), which, again, show PTIC film and PBDB-TF:PTIC blend possess superior photostability. Note that the light spectra we employed in this study contains high portion of ultraviolet photons (250-380 nm), which is the common source of instability of organic materials that associate chemical bond dissociation under high energy photon irradiance (Adv. Energy Mater. 2011, 1, 491; Adv. Energy Mater. 2019, 9, 1902124). The photochemical degradation of conjugated polymer, as electron donors, would involve a chain radical oxidation process, similar to that of P3HT (Thin Solid Films, 2010, 518, 7113).

Fig. 3 | Molecular stacking of PTIC in single crystal. **a**, Schematic illustration of PTIC with planar unfused structure adapts dense packing in crystals that prevents PTIC from photochemical reactions (ICT: intramolecular charge transfer). Single crystal structure and stacking model of

PTIC. **b**, top view and **c**, side view.

3. After the illumination treatment, do the degraded fragments volatilized? The optical density is nearly zero.

Response: Thanks for the suggestions. The loss of optical density is due to the photo-degradation/bleaching of organic semiconductors (J. Mater. Chem. A, 2019, 7, 25088; Adv. Mater., 2020, 32, e2003471). Under the harsh light ageing in open air, photochemical reaction of organic materials involves the photo-oxidation of bonds, bleaching of chromophore group, as well as chemical bond dissociation. It eventually leads to fragmentation of molecules under the severe photochemical reactions, leaving few residues on substrates. We have checked the degraded IT-4F films (to near zero optical density) through deuterated solvent rinsing of substrates. NMR measurements have indicated there almost no characteristic signals of IT-4F.

Review Only Figure ¹H NMR of samples obtained from dichlorobenzene-d₄ rinsing the photon-aged IT-4F film (after 220 h continues one sun-equivalent illumination) (the solvent residues signals of 7.07 ppm are omitted for clarity).

4. About the conclusion, I think it's better to point out there may be other reasons that affecting the chemical stability of these materials as the studied systems are quite limited when compared

to the boom of OSC materials.

Response: Thanks for the suggestions. We have revised the related contents in main text:

Abstract:

Photo-degradation of organic semiconductors remains as an obstacle preventing their durable practice in optoelectronics. Herein, we disclose that volume-conserving photoisomerization of a unique series of acceptor-donor-acceptor (A-D-A) non-fullerene acceptors (NFAs) acts as a surrogate towards their subsequent photochemical reaction. Among A-D-A NFAs with fused, semi-fused and non-fused backbones, fully non-fused PTIC, representing one of rare existing samples, exhibits not only excellent photochemical tolerance in aerobic condition, but also efficient performance in solar cells. Along with a series of in-depth investigations, we identify that the structural confinement to inhibit photoisomerization of these unique A-D-A NFAs from molecular level to macroscopic condensed solid helps enhancing the photochemical stabilities of molecules, as well as the corresponding OSCs. Although other reasons associating with the photostabilities of molecules and devices should not be excluded, we believe this work provides helpful structure-property information toward new design of stable and efficient photovoltaic molecules and solar cells.

Main text:

Even though, there should not exclude other reasons associating on the photostabilities of non-fullerene acceptors and solar cells. We believe the structure-property correlation from these unique examples provides new insights on designing efficient and stable organic semiconductors and the derived solar cells.

Conclusion part:

In this regard, the structural factors, including hinder outward-chain and planar sp^3 carbon-free backbones play important roles to enhance the intrinsic photo-stabilities of these A-D-A NFAs and their derived OSCs. Even though, there are still some other complicate reasons associating on the photostabilities of non-fullerene solar cells. We believe the structure-property correlation from these unique examples can be beneficial to community, providing new chemistry insights for designing photostable organic semiconductors and optoelectronics.

5. I think the authors should update the references as the field is growing very fast. It's better to conclude the latest progress reporting record efficiencies, such as 17-18% efficiencies obtained using non-fullerene acceptors like bo-4cl, eC9.

Response: Thanks for suggestions. Serval reference close to this work has been cited into reference and discussed in main text.

Reference that discuss the photochemical degradation of conjugated polymers and blends (Adv. Energy Mater. 2011, 1, 491; Adv. Energy Mater. 2019, 9, 1902124; Thin Solid Films, 2010, 518, 7113). Works report the up-o-date electron acceptors, BO-4Cl with record efficiency (Natl. Sci. Rev. 7, 1239-1246 (2020)). The latest progresses report on the lifetime study of non-fullerene solar cells (CCS Chem. doi: 10.31635/ccschem.31021.202100852).